# Gene Expression and Mutational Profile in BAP-1 Inactivated Melanocytic Lesions of Progressive Malignancy from a Patient with Multiple Lesions

**DOI:** 10.3390/genes13010010

**Published:** 2021-12-22

**Authors:** Yan Zhou, Andrew C. Nelson, Yuyu He, Sarah A. Munro, Kyu Young Song, Evidio Domingo-Musibay, Alessio Giubellino

**Affiliations:** 1Department of Laboratory Medicine and Pathology, University of Minnesota, Minneapolis, MN 55455, USA; zhou1237@umn.edu (Y.Z.); nels2055@umn.edu (A.C.N.); hexxx876@umn.edu (Y.H.); songx047@umn.edu (K.Y.S.); 2Masonic Cancer Center, University of Minnesota, Minneapolis, MN 55455, USA; musib024@umn.edu; 3Minnesota Supercomputing Institute, University of Minnesota, Minneapolis, MN 55455, USA; smunro@umn.edu; 4Department of Medicine, University of Minnesota, Minneapolis, MN 55455, USA

**Keywords:** BAP1, melanocytic lesions, melanoma, UV radiation response, MAPK, receptor protein kinase

## Abstract

BAP-1 (BRCA1-associated protein 1) inactivated melanocytic lesions are a group of familial or sporadic lesions with unique histology and molecular features. They are of great clinical interest, at least in part due to the potential for malignant transformation and association with a familial cancer predisposition syndrome. Here, we describe a patient with multiple spatially and temporally distinct melanocytic lesions with loss of BAP1 expression by immunohistochemistry. RNA sequencing was performed on three independent lesions spanning the morphologic spectrum: a benign nevus, an atypical tumor, and a melanoma arising from a pre-existing BAP1-inactivated nevus. The three lesions demonstrated largely distinct gene expression and mutational profiles. Gene expression analysis revealed that genes involved in receptor protein kinase pathways were progressively upregulated from nevus to melanoma. Moreover, a clear enrichment of genes regulated in response to UV radiation was found in the melanoma from this patient, as well as upregulation of MAPK pathway-related genes and several transcription factors related to melanomagenesis.

## 1. Introduction

*BAP1* (BRCA1-associated protein 1) gene, located on chromosome 3p21, is a tumor suppressor gene that encodes a deubiquitination enzyme regulating several key cellular pathways [1]. Inherited germline inactivating mutations in BAP1 have recently been found associated with a cancer predisposition syndrome, initially described in two unrelated families by Wiesner et al. [2]. It is characterized by the occurrence of multiple epithelioid melanocytic neoplasms resembling Spitz nevi and increased susceptibility for developing several malignancies in the affected individuals, including uveal melanoma, cutaneous melanoma, renal cell carcinoma, mesothelioma, and other tumors [3,4]. A variety of mutations in both coding and noncoding regions throughout the *BAP1* gene has been identified in the melanocytic lesions and can impair its protein function [5]. Interestingly these lesions harbor a *BRAF* V600E mutation, and characterization of this feature has helped distinguish them from Spitz lesions, which do not have this mutation in their genetic background [2,3,6]. Furthermore, in lesions with combined morphologies, *BRAF* V600E was found in all melanocytes, whereas *BAP1* mutation was restricted to the epithelioid cells, suggesting that the *BAP1*-inactivated melanocytic tumors might arise from common acquired nevi [2,7]. 

Besides presenting as a part of the familial cancer syndrome, *BAP1*-inactivated melanocytic tumors, including benign-appearing nevi, atypical tumors, and melanomas can also occur in a sporadic fashion [8,9]. For both the familial and sporadic lesions, prior studies found that only a minority of the *BAP1*-inactivated melanocytic lesions progress to melanoma, suggesting a relatively low malignant potential [3,7,10]. For an isolated lesion, a conservative complete excision with close clinical follow-up is the standard of care. When multiple occurrences of such melanocytic lesions are present in the same individual, genetic counseling and testing for germline *BAP1* mutation are recommended. However, a considerable number of patients who presented with multiple cutaneous BAP1-inactivated melanocytic lesions had no prior history of *BAP1*-associated malignancies [5], and long-term follow-up is usually recommended. 

While many cases of *BAP1*-inactivated melanocytic tumors have been reported to date in the literature, data on the gene expression profile of these lesions is limited, and little we know about the differential expression of genes of progressively malignant lesions in this category of melanocytic neoplasms. To explore the expression profiles of a full spectrum of *BAP1*-inactivated lesions, here we performed RNA sequencing on three lesions from a young patient with a nevus, an atypical tumor, and a melanoma with BAP1 loss. We also compared the gene expression profiles among the lesions to explore potential markers of tumor progression.

## 2. Materials and Methods

One melanocytic nevus, one atypical tumor, and a melanoma, all with loss of BAP1 expression, from the same patient were selected for subsequent RNA extraction and sequencing studies. Histologic diagnoses of these lesions were confirmed by an expert second opinion.

### 2.1. RNA Sequencing

Unstained slides prepared from the archival paraffin-embedded tissue of the three specimens were macro-dissected to obtain RNA from lesional tissue. Total RNA was extracted and purified using an RNeasy FFPE Kit (Qiagen, Germantown, MD, USA). RNA samples were then quantified and analyzed for quality (Agilent RNA 6000 Nano Kit, Agilent, Santa Clara, CA, US). Library preparation and targeted gene enrichment were performed with the TruSight RNA Pan-Cancer Panel Kit according to the manufacturer’s protocol (Illumina, San Diego, CA, USA). Libraries were sequenced on the Illumina NextSeq 550 System. FASTQ file analysis was performed using the Illumina BaseSpace RNA-Seq Alignment Application 2.0.2. Gene-level counts [11] were created using this pipeline developed and supported by Illumina. 

Annovar [12] (15 April 2018 version) was used to annotate the variant call files with clinical genomic information, including gnomAD minor allele fractions, COSMIC cancer listings, and NCBI ClinVar clinical significance. Variants that meet the following criteria were kept: (1) alternate allele depth (AD) > 5 and VAF ≥ 5% or (2) AD = 4 or 5 and VAF ≥ 15%. Plus, we excluded small indels in repetitive sequence with VAF < 10%. Annotated and quality-filtered variant calls were reviewed by a board-certified molecular pathologist for potential or known clinical significance using established professional guidelines [13]. 

### 2.2. RNA Expression and Pathway Analysis

Gene-level counts from the BaseSpace RNA-Seq Alignment Application were analyzed using custom R scripts and open-source R packages. Any genes that did not have at least 1 cpm (count per million) were removed. A 1.5-fold cutoff of log2 cpm values was used to evaluate gene expression differences between any of the two samples. Several approaches were used for pathway analysis based on all the differentially expressed genes: GO term enrichment analysis, ToppGene Suite [14] and EnrichR [15,16,17] for functional enrichment analysis, and Gene Set Enrichment Analysis (GSEA) [18].

## 3. Results

### 3.1. Case Presentation

A 31-year-old man presented with a history of multiple atypical melanocytic lesions, biopsied at an outside institution. He first presented subsequently at our hospital with an ear lesion in the right superior helix with the clinical appearance of a papule, which was tender. On histology, there was a severely atypical compound melanocytic proliferation with abundant pagetoid scatter of single melanocytes across the breath of the epidermis and multiple dermal mitoses (mitotic index of 5 per mm^2^) (Figure 1A–C). The ki67 stain demonstrated a modest increase in the labeling of dermal melanocytes, while p16 expression was lost in the atypical melanocytes, which suggested *CDKN2A* genomic loss. BAP1 immunohistochemistry demonstrated a loss in part of the lesion. Altogether a diagnosis was rendered of melanoma developing in the context of a melanocytic nevus with associated *BAP1* genomic loss (melanoma ex-“BAPoma”). The Breslow thickness was 0.75 mm with no ulceration, no vascular invasion, and no regression, placing the pathologic staging as a pT1a lesion.

Another lesion was detected one year later in the right posterior ear, presenting clinically as a 2 mm pink pearly papule. Histologically the lesion presented as a mostly intradermal proliferation of large, oval melanocytes with abundant pale cytoplasm and large but monomorphous vesicular nuclei and small nucleoli, sparse mitoses, and with loss of BAP1 expression (Figure 1D–F). A final diagnosis of atypical Spitz nevus with *BAP1* loss was rendered.

The following year, other melanocytic lesions were removed, including a lesion in the posterior neck with a clinical appearance of a dome-shaped lesion. On histology, the lesion presented as a dermal melanocytic proliferation composed of a regular melanocytic nevus in conjunction with more ovoid melanocytes with *BAP1* loss (Figure 1G–I). Given the histologic and immunohistochemical features, this lesion was classified as a BAP-1 inactivated nevus (“BAPoma”).

### 3.2. Gene Expression and Mutation Analyses

We performed a pairwise exploration of the relative differences in gene expression levels between these lesions (Figure 2A–C). A total of 40 upregulated and 53 downregulated genes were identified in the atypical tumor compared to the *BAP1*-inactivated nevus, 26 upregulated and 106 downregulated genes in the melanoma compared to the atypical tumor, and finally, 51 upregulated and 158 downregulated genes in the melanoma compared to the BAP1-inactivated nevus. Among the genes upregulated in the atypical tumor compared to the nevus were: *SOX10*, the receptor tyrosine kinases *ROS1* and *NTRK3*, *PLA2G2A*, and *HOXA11*, while *DUSP2*, *PDGFD*, and *ELN* were downregulated (Figure 2A). *PAX3*, a transcription factor involved in melanocyte development, the receptor tyrosine kinase *KIT*, *CDKN1C*, and *EPHA5* were upregulated in the melanoma compared to the atypical nevus, while *NTRK3* and *ROS1* were downregulated in the same comparison (Figure 2B). Finally, Figure 2C illustrates a volcano plot of the comparison of the melanoma versus the nevus, with notable upregulation of *KIT*, *PAX3*, *SOX10*, *CDKN1C*, and *DUSP9*, and downregulation of *ATF3* and *DUSP2*.

By analyzing and comparing the relative expression levels of the top 50 most variable genes among the three lesions, we identified a group of differentially expressed genes (Appendix A). We then wanted to explore which genes are progressively upregulated or downregulated from a benign *BAP1*-inactivated nevus to a *BAP1*-inactivated melanoma. Figure 3 illustrates this concept. Overall, genes that are progressively upregulated from nevus to atypical tumor to melanoma include *SOX10*, a transcription factor important in melanocytic differentiation, the protein tyrosine kinases *IGF1R*, *KIT* (the latter mutated in acral melanoma, mucosal melanoma, and melanoma of chronically sun-damaged skin) and *EPHA5*, the transcription factor *PAX3* (involved in melanocyte development), the Dual Specificity Phosphatase *DUSP9*, *WNT11* and *IRF4/MUM1* (a gene regulated by MITF in melanocytic cells). In contrast, genes that are progressively downregulated from nevus to atypical tumor to melanoma include *NCAM1* (implicated in cell-cell adhesion and reportedly downregulated in several human cancer, suggesting a tumor repressor role), *HSPA1A*, *DKK1* (a secreted inhibitor of the β-catenin dependent Wnt signaling pathway and involved in induction of cancer evasion of immune surveillance), the growth factor *PDGFD*, the protein phosphatase *DUSP2* (involved in the negative regulation of members of the mitogen-activated protein kinase (MAPK) superfamily), and *LRP1B* (frequently mutated in melanoma) [19].

In order to understand the collective functions and the cellular molecular pathways related to the differentially expressed genes in our samples, we performed gene set enrichment analysis. Using the Hallmark gene sets within the EnrichR software analysis, when comparing the atypical tumor with the nevus, there was a clear modulation of the epithelial-mesenchymal transition pathway, as well as upregulation of the KRAS signaling pathway and interleukin/STAT signaling, and downregulation of TNF-α signaling. When comparing the melanoma with both the nevus and the atypical tumor, it was interesting to observe regulation of the UV response, particularly important in melanomagenesis. Specifically, genes upregulated or downregulated in response to UV radiation involve *CDKN1C*, *CDK2*, *COL11A1*, *KIT*, and *IGF1R*, the majority of which are identified as differentially upregulated in melanoma versus atypical tumor or nevus (Figure 2B,C and Figure 3A). In this comparison, there was also the regulation of TNF-α signaling and epithelial-mesenchymal transition (Figure 4A,B).

GO term enrichment analysis was also performed and showed that expression profiles involving the extracellular region and component of the membrane were enriched progressively from nevus to atypical tumor to melanoma (Appendix A). Particularly, pathways related to protein kinase activity in general as well MAPK activity and protein tyrosine kinase activity were significantly upregulated in both the atypical tumor and melanoma compared to the nevus (Appendix A). 

Genes harboring pathogenic or likely pathogenic mutations were heterogeneous across the three lesions (Table 1). *NIPBL*, *AKAP9*, and *EP400* were mutated in both the nevus and atypical tumor, although the specific mutations were different between the two lesions. In addition, mutations in *PDGFRB*, *HIF1A*, and *MAP2K1* genes were identified in the nevus, and mutations in *HDAC2*, *PRK2*, *ARID2*, *BRCA2*, *NOTCH3*, *NOTCH2*, *BRAF*, *CSF1R*, and *PTEN*, among others, were found in the atypical nevus. To be noted, *BRAF* V600E was identified in both atypical tumors and melanoma. The mutation in *BAP1* was captured in the melanoma, as well as mutations in *KMT2A* (a known cancer driver mutation in melanoma) and *TIAM1* (a gene involved in the RAC1 signaling pathway affecting cell shape and migration).

## 4. Discussion

*BAP1*-inactivated melanocytic lesions exist in a spectrum that goes from benign melanocytic nevi, usually combined (“BAPomas”), to intermediate lesions with atypical features (akin to the atypical Spitz tumor) and ending with malignant melanoma. While initially categorized within the Spitz family of melanocytic lesions, genetic analysis has revealed a molecular profile, including the presence of *BRAF* mutation, that is not compatible with this classification. Thus, in the recent WHO classification, these lesions have been classified separately as combined melanocytic lesions with a distinct morphologic and molecular signature (*BAP1* inactivation and *BRAF* V600E mutation). From their first description in the seminal paper by Wiesner et al. [2], there has been a lot of interest in these lesions as a paradigm for newly recognized melanocytic lesions with distinct histologic morphology and a well-defined molecular signature. The last WHO classification of cutaneous tumors classifies *BAP1*-inactivated lesions in a separate category, recognizing its unicity. However, besides the pathognomonic mutation, little is known regarding the gene expression profile and other gene mutations occurring in these lesions. To our knowledge, this is the first study to compare gene expression levels in a series of *BAP1*-inactivated melanocytic lesions of progressive malignancy, from nevus to atypical tumor to melanoma in a single patient. 

The same WHO classification released in 2018 put an accent on the role of UV-related damage in melanocytic tumorigenesis by dividing the most common melanocytic lesions in those related to low cumulative solar damage (CSD) and high CSD. *BAP1* mutation may occur as the driver genetic event in low CSD melanoma, but there is not a specific relation with sun—damage reported for the new category of *BAP1*-inactivated lesions. In our study, we found a clear enrichment of genes regulated in response to UV radiation, based on the Molecular Signature Database (MSigDB), suggesting that sun damage is a relevant step in the multistep process of tumor progression in these lesions. 

Our study possesses potential clinical significance by suggesting multiple therapeutic targets in patients with *BAP1*-inactivated melanoma, the treatment for whom is otherwise limited to conventional therapies. These targets may involve genes in protein kinase pathways or governing transcriptional regulation, as discussed below. Based on our analysis, activation of protein kinase pathways, especially the MAPK pathway, is largely observed in the atypical tumor and melanoma arising from *BAP1*-inactivated nevus compared to nevus, consistent with current knowledge of melanocytic progression. MAPK activation represents a common aberrant signaling pathway in cutaneous melanoma associated with a wide variety of somatic genetic alterations. This knowledge has resulted in effective targeted therapies targeting this pathway; the success of BRAF inhibitors alone or in combination with MEK inhibitors [20] is a testament to the importance of molecular and mechanistic studies to understand the pathogenesis and discover relevant targets for cancer therapy. Thus, our findings support the use of similar therapeutic regimens in our patient as adjuvant treatment. To further support this, we found downregulation of the protein phosphatase *DUSP2*, involved in the negative regulation of members of the MAPK superfamily, and upregulation of the KRAS signaling pathway. In addition, genes encoding protein tyrosine kinases, *KIT* and *EPHA5*, relevant receptors upstream of the MAPK pathway, are particularly highly expressed in the melanoma compared to *BAP1*-inactivated nevus. Interestingly, a mutation in the *MAP2K1* gene was identified in the BAPoma from our patient; similar mutations in this gene have been reported in a large spectrum of melanocytic lesions, including BAPomas [21].

IGF1R (insulin-like growth factor I receptor), another receptor tyrosine kinases found to be upregulated in our sample, has been found to have anti-apoptotic properties and to be overexpressed in multiple cancers including melanoma [22], where may represent a potential therapeutic target. 

Among the protein kinase activation pathways, the expression levels of *ROS1* and *NTRK3* are only upregulated in the atypical tumor compared to the nevus and melanoma. Interestingly, various kinase fusions, including *ROS1* and *NTRK3*, were found in Spitz melanocytic lesions [23]. These various fusion events are mostly mutually exclusive and are believed to be an early event in tumorigenesis [24], but do not necessarily confer malignant potential and are actually present often in benign or intermediate lesions, but rarely in melanoma. 

Another interesting group of upregulated genes in the atypical tumor and the melanoma is related to sequence-specific DNA binding and regulatory region nucleic acid binding, specifically transcriptional regulation. This gene category includes *HOXA11*, *SOX10*, *ETV5*, *IRF4*, *PAX3*, *PRDM7*, *TFAP2B*, and *WNT11*. Among those, the transcription factor SOX10 has been shown to be involved in melanomagenesis in animal models [25,26] and, more recently, in the regulation of melanoma cell invasion, through the regulation of melanoma inhibitory activity (MIA) expression [27]; thus SOX10, besides its role as a marker of melanocytic differentiation in diagnostic pathology, represents a potential therapeutic target for melanoma. SOX10 was also found to regulate immunogenicity in melanoma through IRF4 (interferon regulatory factor 4)/MUM1 [28], another transcription factor discovered in our dataset. *IRF4* is a gene regulated by MITF in melanocytic cells, and it was also found to have a functional variant associated with increased Breslow thickness, conferring a worse survival in melanoma [29]. *WNT11* was shown to play an important role in neural crest migration and appears to have a role in the aberrant activation of Wnt signaling in melanoma. Finally, *PAX3* (paired box gene 3 transcription factor) was shown, in conjunction with the transcription factor *ETS1*, to promote melanoma cells proliferation and metastasis by increasing the expression of MET, the HGF receptor [30]. Altogether, these findings highlight the role of several transcription factors as critical players in melanoma. Given their role as focal and convergence points of several signaling pathways and their role in tumor progression and resistance to therapy [31], these transcription factors are potential targets to explore for the development of future therapy for melanoma, including those arising in the background of *BAP1* inactivation.

There are some limitations in our study that we need to acknowledge. First, only three samples from the same patient were available at the time of the experiment; thus, these data represent a single set of lesions, and future studies on multiple samples from several patients will be needed to generalize any of the biomarkers identified. Second, only a panel of 1400 cancer-related genes were analyzed, and the source material was formalin-fixed and paraffin-embedded, a challenging source that is prone to RNA fragmentation impacting the number of genes that we were able to capture with our assay. Future experiments on a larger panel of genes or the entire transcriptome may reveal additional significant markers of tumor progression in this type of lesion.

## 5. Conclusions

In conclusion, our study suggests that a *BAP1*-inactivated nevus, an atypical tumor, and melanoma with the same background from the same patient possess distinct gene expression profiles, with notable upregulation of genes involved in protein kinase pathways in the progression from nevus to melanoma. Moreover, we found a clear enrichment of genes regulated in response to UV radiation in the melanoma from this patient, as well as upregulation of MAPK pathway-related genes and several transcription factors related to melanomagenesis. Current treatments for patients with *BAP1*-inactivated melanoma are limited to conventional therapies used in other more common melanomas, which may not address the molecular mechanisms of these lesions entirely. Thus, our study may suggest potential novel targets for personalized therapy in these patients.

## Figures and Tables

**Figure 1 genes-13-00010-f001:**
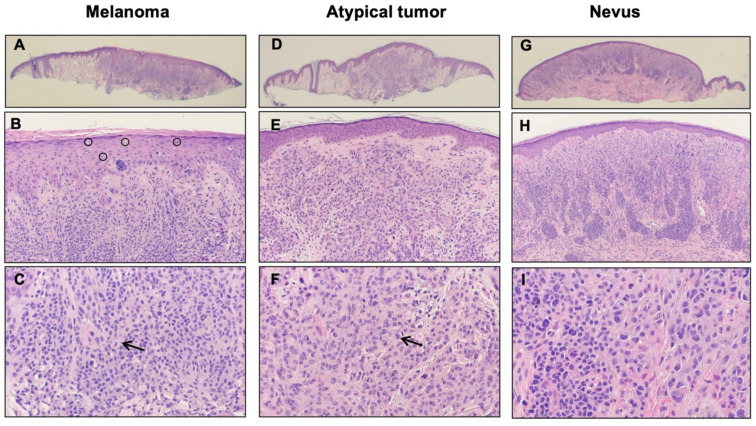
Histopathology of three representative lesions. (**A**–**C**) Melanoma; an atypical compound melanocytic proliferation with a nevoid appearance at low power (**A**), with abundant pagetoid scatter of single melanocytes within the epidermis ((**B**), circles) and multiple dermal mitoses ((**C**), arrow); (**D**–**F**) atypical tumor; a predominantly dermal melanocytic proliferation of atypical yet monomorphous large oval melanocytes with occasional mitoses ((**F**), arrow); (**G**–**I**) nevus; a dermal proliferation of small regular melanocytes in conjunction with large ovoid melanocytes (**I**). Circles (**B**): pagetoid spread of the tumor cells. Arrows (**C**,**F**): mitoses.

**Figure 2 genes-13-00010-f002:**
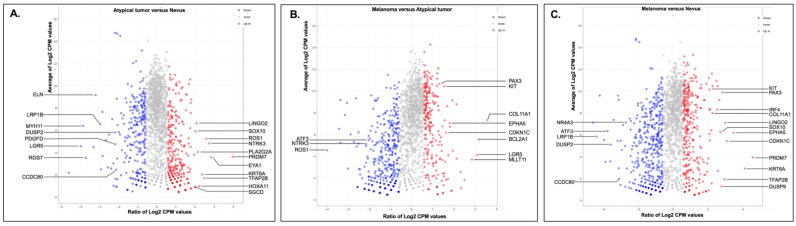
The MA plots showing upregulation and downregulation of the genes between each of the two BAP1-inactivated melanocytic lesions. (**A**) Atypical tumor versus Nevus; (**B**) Melanoma versus Atypical tumor; (**C**) Melanoma versus Nevus.

**Figure 3 genes-13-00010-f003:**
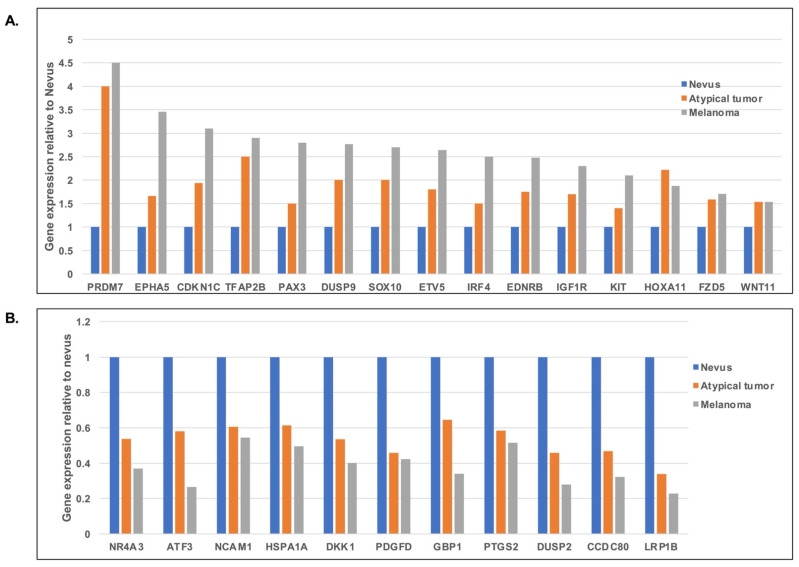
(**A**) Relative expression levels of the genes showing the upregulated trend of expression from nevus, atypical tumor to melanoma. (**B**) Relative expression levels of the genes showing the downregulated trend of expression from nevus, atypical tumor to melanoma.

**Figure 4 genes-13-00010-f004:**
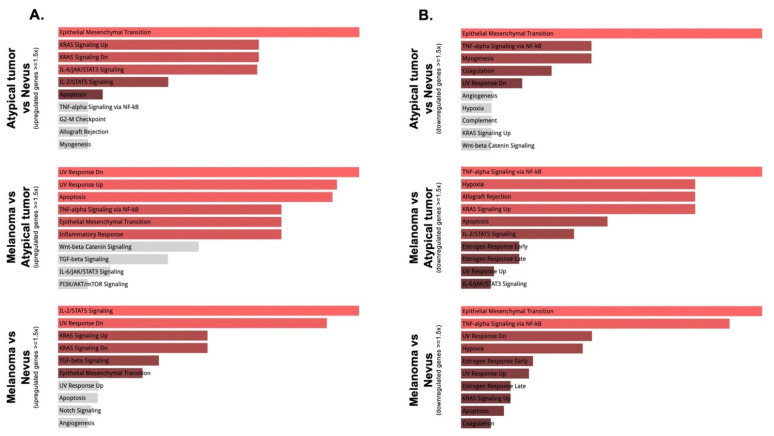
Gene enrichment analysis using the EnrichR platform. Upregulated (**A**) and downregulated (**B**) pathways (in the Hallmark category) in the following comparisons: Atypical tumor versus Nevus; Melanoma versus Atypical tumor; and Melanoma versus Nevus. Statistical significantly enriched terms are displayed in red, while gray bars are not statistically significant.

**Table 1 genes-13-00010-t001:** List of genes with identified pathogenic or likely pathogenic mutations in the BAP-1 inactivated nevus, atypical tumor, and melanoma.

Lesion	Gene	Mutation	Type
Nevus	*NIPBL*	p.Q338X	stopgain
	*AKAP9*	p.Q2911X	stopgain
	*EP400*	p.E2200X	stopgain
	*CREBBP*	p.Q1075X	stopgain
	*COL1A1*	p.P817fs	frameshift deletion
	*ZNF687*	p.Q474X	stopgain
	*PDGFRB*	p.Q412X	stopgain
	*ASPH*	p.R659X	stopgain
	*NT5C2*	p.Q173X	stopgain
	*NIN*	p.Q1291X	stopgain
	*HIF1A*	p.Q379X	stopgain
	*CHD2*	p.Q366X	stopgain
	*USP7*	p.Q822X	stopgain
	*MYO18A*	p.Q830X	stopgain
	*KPNB1*	p.Q111X	stopgain
	*RPS6KA3*	p.R383W	nonsynonymous SNV
	*TBL1XR1*	p.S332F	nonsynonymous SNV
	*MAP2K1*	p.A172V	nonsynonymous SNV
Atypical tumor	*NIPBL*	p.Q1567X	stopgain
	*AKAP9*	p.Q2480X	stopgain
	*EP400*	p.Q148X	stopgain
	*CAD*	p.Q30X	stopgain
	*PPP1CB*	p.Q293X	stopgain
	*BIRC6*	p.Q1072X	stopgain
	*COL6A3*	p.Q2366X	stopgain
	*TFG*	p.Q266X	stopgain
	*EIF4A2*	p.Q209X	stopgain
	*AFF4*	p.Q537X	stopgain
	*ARHGAP26*	p.R120X	stopgain
	*MAML1*	p.Q683X	stopgain
	*DST*	p.Q1890X	stopgain
	*HDAC2*	p.Q128X	stopgain
	*PTK2*	p.Q734X	stopgain
	*SYK*	p.Q239X	stopgain
	*NUMA1*	p.Q832X	stopgain
	*PICALM*	p.Q256X	stopgain
	*EED*	p.Q302X	stopgain
	*SIK3*	p.Q675X	stopgain
	*KDM5A*	p.R266X	stopgain
	*PRICKLE1*	p.Q348X	stopgain
	*ARID2*	p.Q720X	stopgain
	*NUP107*	p.Q236X	stopgain
	*BRCA2*	p.Q2943X	stopgain
	*TCF12*	p.Q605X	stopgain
	*TOP2A*	p.Q517X	stopgain
	*ITGB3*	p.Q616X	stopgain
	*SMARCA4*	p.R978X	stopgain
	*NOTCH3*	p.Q646X	stopgain
	*CSNK2A1*	p.Q71X	stopgain
	*NCOA3*	p.Q478X	stopgain
	*ETS2*	p.Q234X	stopgain
	*EP300*	p.Q523X	stopgain
	*OFD1*	p.Q39X	stopgain
	*AR*	p.Q825X	stopgain
	*ZMYM3*	p.Q763X	stopgain
	*BMPR1A*	p.R244X	stopgain
	*NOTCH2*	p.Q1814X	stopgain
	*TPR*	p.Q2344X	stopgain
	*BRAF*	p.V600E	nonsynonymous SNV
	*CSF1R*	p.A781V	nonsynonymous SNV
	*PTEN*	p.T277I	nonsynonymous SNV
Melanoma	*LRPPRC*	p.G1050fs	frameshift insertion
	*EXT2*	p.R215X	stopgain
	*SPEN*	p.Q3373X	stopgain
	*BAP1*	p.Y223X	stopgain
	*AHR*	p.Q705X	stopgain
	*KMT2A*	p.Q1207X	stopgain
	*CDH1*	p.Q388X	stopgain
	*RABEP1*	p.Q225X	stopgain
	*CLTC*	p.Q1358X	stopgain
	*SUGP2*	p.R831X	stopgain
	*TIAM1*	p.Q714X	stopgain
	*BRAF*	p.V600E	nonsynonymous SNV

## Data Availability

The data presented in this study are available in the Appendix A and also upon request from the corresponding author.

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
