# Peer review of "Gene Expression and Mutational Profile in BAP-1 Inactivated Melanocytic Lesions of Progressive Malignancy from a Patient with Multiple Lesions"

_genes, 2021, doi:10.3390/genes13010010_

Round 1

Reviewer 1 Report

The paper from Dr. Zhou et al. is an interesting study including comparisons re. gene expression in three lesions from the same patient but all characterized by BAP1-inactivation. The paper offers insights on the tumorigenesis involving BAP1-inactivated lesions, from a nevus to a melanoma. As authors clearly show, these are peculiar melanocytic lesions and potential clinical sign of the recently described BAP1-TPDS.

The limitations of the study are self-evident (three lesions, one patients), nevertheless the results can be the basis to wider studies on the specific mechanisms of tumorigenesis of BAPomas.

I have some minor suggestions, hence not mandatory.

1) In the abstract, use predisposition instead of disposition

2) Maybe the references can be more up-to-date citing also the paper from BIG consortium (Comprehensive Study of the Clinical Phenotype of Germline BAP1 Variant-Carrying Families Worldwide), i.e. in line 35 or wherever the authors prefer (the clinical spectrum of BAP1-TPDS is mostly but not conclusively defined)

3) While the graphs are self explaining, I'm not sure that for a non-expert reader the genes involved in UV response are clearly highlighted. Maybe some notes can be added on line 166 or in supplemental table 1.

Author Response

Responses to Reviewer 1

1) In the abstract, use predisposition instead of disposition.

Response: Thank you for the suggestion, we have made the change in the revised manuscript (line 14).

2) Maybe the references can be more up-to-date citing also the paper from BIG consortium (Comprehensive Study of the Clinical Phenotype of Germline BAP1 Variant-Carrying Families Worldwide), i.e. in line 35 or wherever the authors prefer (the clinical spectrum of BAP1-TPDS is mostly but not conclusively defined)

Response: The suggested reference 4 was included in line 35, and the numbering of references were updated throughout the manuscript.

3) While the graphs are self-explaining, I'm not sure that for a non-expert reader the genes involved in UV response are clearly highlighted. Maybe some notes can be added on line 166 or in supplemental table 1.

Response: Genes that are differentially expressed among the lesions and involved in the UV response were highlighted in lines 198-201. In addition, typos in Figure 2B were corrected in the updated Figure.

Additional changes include:

  1. revision of the figure 1 caption to include more details (lines 137-141).
  2. Minor edits on the spellings of words or phrases throughout the manuscript.

Reviewer 2 Report

This an interesting article investigating gene expression and mutational profile in BAP-1 inactivated melanocytic lesions of progressive malignancy from a patient with multiple lesions . The artcle is well written and conclusions are consistent with experimental data. In the Discussion section the clinical translational impact of the study should be better specified.

Author Response

Responses to Reviewer 2

1)This an interesting article investigating gene expression and mutational profile in BAP-1 inactivated melanocytic lesions of progressive malignancy from a patient with multiple lesions. The article is well written and conclusions are consistent with experimental data. In the Discussion section the clinical translational impact of the study should be better specified.

Response: Thank you for your comments. We highlighted and summarized the potential clinical significance of our study in lines 259-262.

Additional changes include:

  1. revision of the figure 1 caption to include more details (lines 137-141).
  2. Minor edits on the spellings of words or phrases throughout the manuscript.
